# Clinical Markers of Panic and Generalized Anxiety Disorder: Overlapping Symptoms, Different Course and Outcome

**DOI:** 10.3390/jpm13030491

**Published:** 2023-03-09

**Authors:** Alice Caldiroli, Lia Colzani, Enrico Capuzzi, Cecilia Quitadamo, Davide La Tegola, Teresa Surace, Stefania Russo, Mauro Capetti, Silvia Leo, Agnese Tringali, Matteo Marcatili, Francesco Zanelli Quarantini, Fabrizia Colmegna, Antonios Dakanalis, Massimiliano Buoli, Massimo Clerici

**Affiliations:** 1Department of Mental Health and Addiction, Fondazione IRCCS San Gerardo dei Tintori, Via G.B. Pergolesi 33, 20900 Monza, Italy; 2Department of Medicine and Surgery, University of Milan Bicocca, Via Cadore 38, 20900 Monza, Italy; 3Department of Medicine and Surgery, University of Milan, Via Festa del Perdono 7, 20122 Milan, Italy; 4Department of Neurosciences and Mental Health, Fondazione IRCCS Ca’ Granda Ospedale Maggiore Policlinico, 20122 Milan, Italy; 5Department of Pathophysiology and Transplantation, University of Milan, 20122 Milan, Italy

**Keywords:** generalized anxiety disorder, panic disorder, comorbidities, chronicity, outcome, anxiety disorders

## Abstract

Generalized Anxiety Disorder (GAD) and Panic Disorder (PD) share underlying neurobiological mechanisms and several clinical features which, with medical comorbidities, may increase misdiagnosis and delay proper treatment. The aim of the study was to evaluate the association between clinical/socio-demographic markers and GAD/PD diagnosis. Outpatients (N = 290) with PD or GAD were identified in mental health services in Monza and Milan (Italy). Descriptive analyses and a binary logistic regression model were performed. Post-onset psychiatric (*p* = 0.05) and medical (*p* = 0.02) multiple co-morbidities were associated with GAD; treatment with selective serotonin reuptake inhibitors (SSRIs) was associated with PD, while GAD diagnosis was associated with treatment with atypical antipsychotics or GABAergic drugs (*p* = 0.03), as well as psychodynamic psychotherapy (*p* < 0.01). Discontinuation of the last pharmacological treatment was associated with GAD diagnosis rather than the PD one (*p* = 0.02). GAD patients may have a worse prognosis than PD patients because of more frequent multiple co-morbidities, relapses and poorer treatment compliance. The different treatment approaches were consistent with the available literature, while the association between GAD and psychodynamic psychotherapy is an original finding of our study. Further studies on larger samples are necessary to better characterize clinical factors associated with GAD or PD.

## 1. Introduction

According to epidemiological surveys, anxiety disorders affect roughly one-third of the global population [1] in their lifetime and are the most common mental illnesses worldwide [2]. From an economic point of view, anxiety disorders are associated with immense health care costs as a result of medical assistance and social impairment including workday loss [3]. For instance, some authors have demonstrated that the work loss days for these conditions are higher than for common somatic diseases such as diabetes [4]. Anxiety disorders are characterized by high chronicity, which significantly affects quality of life as well as familiar, social and occupational functioning [2].

The group of anxiety disorders includes generalized anxiety disorder (GAD), panic disorder (PD), phobic disorders and two disorders that are commonly confined to childhood (separation anxiety disorder and selective mutism). GAD and PD are prevalent conditions with a 12-month prevalence of about 4% for the first condition and 2% for the second [1,5]. The diagnostic criteria of GAD and PD first appeared in 1980 with the publication of the third edition of the Diagnostic and Statistical Manual of Mental Disorders (DSM), where the previously defined anxiety neurosis was split into these two disorders [6]. The question of the clinical differences between GAD and PD has always been a topic of interest in the literature, supporting the diagnostic separation of these two disorders. According to DSM-5 criteria, the hallmark of GAD is an excessive, out-of-control anxiety and worry (apprehensive expectation) whilst PD is characterized by recurrent and unexpected panic attacks [7]. However, even though core symptoms of these two conditions are clearly defined, the comorbidity with mood disorders [4], the co-occurrence of medical conditions mimicking anxiety symptoms (e.g., thyroid diseases) [8] or even the presence of patients affected by both GAD and PD hamper correct diagnosis and can delay the prescription of proper treatment [1,9,10]. On the other hand, biomarkers could be helpful in differentiating anxiety disorders and predicting the clinical path of patients affected by these conditions, but research on this topic is very limited with regard to GAD and PD [11]. 

As mentioned above, GAD shares clinical features with PD (e.g., somatic manifestations of anxiety) and the available literature indicates high rates of comorbidity between the two conditions. More than 20% of patients with GAD would also be affected by PD [12,13,14,15]. Regarding neurobiology, both GAD and PD may be the result of dysfunction of specific brain areas such as the amygdala, albeit each of these disorders is clearly the result of complex gene–environment interactions [16]. 

On the other hand, differences in clinical predictors of prognosis were identified in patients affected by GAD compared to those with PD [17]. Individuals with GAD were reported to be older, have an earlier age at onset, longer duration of illness and more psychiatric comorbidity than the patients affected by PD [17,18]. In contrast, individuals with PD seem to be more prone to show substance use disorders compared to patients affected by GAD [17,18]. Despite these preliminary findings, the studies regarding the differences in the clinical path of the two disorders are limited in number [19], even though these data would be useful for personalized medicine and proper management of patients [10,20]. This aspect is complicated by the fact that most available treatments are labelled for both the conditions, without clear evidence of efficacy in one or the other disorder; this is the case, for example, with antidepressants or cognitive behavioral therapy (CBT) [21,22].

In the light of these considerations, the purpose of this study was to identify clinical/socio-demographic differences between GAD and PD and evaluate the association between statistically significant variables and GAD/PD diagnosis. We hypothesize that the identification of potential clinical predictors for GAD or PD may favor an early diagnostic process and may promote personalized medicine.

## 2. Materials and Methods

### 2.1. Sample and Study Design

This is an observational retrospective study. Outpatients suffering from PD or GAD were identified from medical records in the pre-pandemic period (2016–2019) at two mental health services—Fondazione IRCCS San Gerardo dei Tintori (Monza) and Fondazione IRCCS Ca’ Granda, Ospedale Maggiore Policlinico (Milan). Study procedures were reviewed and approved by the local accredited Medical Ethics Review Committee (Area 2 Ethic Committee) of Fondazione IRCCS San Gerardo dei Tintori (Monza). The research project conformed to the provisions of the Declaration of Helsinki regarding medical research in humans in line with good clinical practice requirements.

Inclusion criteria were (1) age 18–65 years; (2) ability and willing to give informed consent; (3) fluency in Italian; (4) diagnosis of GAD or PD according to the criteria of the Diagnostic and Statistical Manual of Mental Disorder, 5th edition [7]. Exclusion criteria were (1) intellectual disability; (2) lack of clinical and socio-demographic information; (3) patients treated for less than three months in the outpatient clinics for the impossibility of a comprehensive collection of clinical variables. In the case of psychiatric comorbidity (e.g., with unipolar depression), GAD or PD was the disorder that affected patients for a longer time and/or was responsible for more current severe symptoms or disability.

The following variables were collected: age, gender, occupation and marital status, age at onset, duration of illness, duration of untreated illness (DUI), family history of psychiatric disorders, multiple family history of psychiatric disorders (yes/no), pre-onset psychiatric comorbidity, pre- and post-onset psychiatric multiple co-morbidities (yes/no), presence and type of personality disorder, pre- and post-onset substance misuse, pre- and post-onset poly-substance misuse (yes/no), pre- and post-onset medical comorbidity, pre- and post-onset medical multiple co-morbidities (yes/no), suicide attempts (yes/no), hospitalizations (yes/no), the presence and type of obstetric complications, main type of pharmacological treatment, duration of treatment, poly-therapy (yes/no), side effects, presence of poly-side effects (yes/no), reason for discontinuation of therapy (rather than for long-term treatment effectiveness), the presence of lifetime psychotherapy (yes/no) and type of psychotherapy.

In the case of multiple visits, the last one was taken into account for the collection of data. DUI was considered as the time between the onset of GAD/PD and the prescription of a proper pharmacological/psychological treatment (labelled antidepressants, pregabalin for GAD, CBT) [9,23]. Suicide attempt was defined as self-harm combined with the intent to die. Self-harm without intent was not taken into account [24].

### 2.2. Statistical Analyses

Descriptive analyses on the total sample were performed. Continuous and qualitative variables were compared between the groups identified according to the diagnosis (GAD or PD) by multivariate analyses of variance (MANOVAs) and χ^2^ tests, respectively. In the case of co-presence of GAD and PD (25 subjects), a patient was assigned to a group according to the current most severe disorder. A binary logistic regression model was then performed considering statistically significant factors from univariate analyses as independent variables and the GAD/PD diagnosis as a dependent one. The goodness of fit of the model was assessed by Omnibus and Hosmer–Lemeshow tests. The significance was set at *p* ≤ 0.05, and confidence intervals at 95% for odds ratios were calculated where applicable.

Statistical Package for Social Sciences (SPSS) for Windows (version 26.0) was used as the statistical program.

## 3. Results

### 3.1. Descriptive Analyses and Diagnostic Group Comparisons

The total sample of patients (N = 290) included 105 males and 185 females; in total, 131 (45.2%) patients were affected by GAD and 159 (54.8%) by PD. The mean age of patients was 45.10 years (±15.39). Demographic and clinical data of the total sample and groups of patients identified by a diagnosis of GAD or PD are summarized in Table 1.

The two groups of patients were statistically different for age (F = 10.37, *p* < 0.01) and age at onset (F = 5.92, *p* = 0.02). GAD patients were older (*p* < 0.01) and had a higher age at onset (*p* = 0.02) than PD ones. Moreover, the two groups of patients showed significant differences in terms of pre-onset medical multiple co-morbidities (χ^2^ = 6.05, OR = 0.47, *p* = 0.01) and post-onset psychiatric (χ^2^ = 5.55, OR = 0.38, *p* = 0.02) and medical (χ^2^ = 12.01, OR = 0.36, *p* < 0.01) multiple co-morbidities (PD < GAD).

Regarding psychopharmacological treatment, PD patients were more frequently treated with selective serotonin reuptake inhibitors (SSRIs) while GAD patients with second generation antipsychotics (SGAs) and GABAergic drugs (χ^2^ = 19.09, *p* = 0.01). Furthermore, patients in the GAD group discontinued treatment more frequently for poor compliance or relapse compared to PD patients (χ^2^ = 10.10, *p* = 0.01). Finally, GAD patients had received lifetime dynamic psychotherapy more frequently than PD ones (χ^2^ = 8.13, *p* = 0.04). No other statistically significant differences were found between the two diagnoses (Table 1).

### 3.2. Binary Logistic Regression Analysis

The goodness-of-fit test (Hosmer and Lemeshow Test: χ^2^ = 2.89, *p* = 0.94) showed that the model including age, age at onset, the presence of post-onset psychiatric multiple co-morbidities, the presence of pre- and post-onset medical multiple co-morbidities, type of main pharmacological treatment, type of psychotherapy and reason for treatment discontinuation (rather than for long-term effectiveness) as independent variables and GAD/PD diagnosis as a dependent one was reliable, allowing for a correct classification of 65.0% of the cases. In addition, the model was overall significant (Omnibus test: χ^2^ = 38.30, *p* < 0.01).

Post-onset psychiatric multiple co-morbidities (*p* = 0.05), post-onset medical multiple co-morbidities (*p* = 0.02) and discontinuation of treatment (rather than for long-term effectiveness) (*p* = 0.02) were associated with GAD diagnosis. Moreover, SSRIs were more frequently prescribed in PD while SGAs or GABAergic drugs (*p* = 0.03) in GAD. Psychodynamic psychotherapy (*p* < 0.01) was more frequently administered in GAD versus PD patients (Table 2).

## 4. Discussion

In this naturalistic retrospective study, we investigated potential clinical and socio-demographic differences between GAD and PD in a sample of 290 outpatients. The present research produced three main results.

The first interesting finding is that patients with GAD were older and had a higher age at onset than those suffering from PD. This result appears consistent with most literature, which reports a later onset of GAD as compared with other anxiety disorders [25]. This observation is supported by the fact that while PD is characterized by a substantial acute autonomic hyperactivity, GAD patients would present a milder and chronic autonomic nervous system dysregulation [26,27]. 

Second, post-onset psychiatric and medical multiple co-morbidities were associated with GAD diagnosis, while pre-onset medical multiple co-morbidities were more frequent in GAD patients than PD ones only in the univariate analyses. Many medical conditions can precede the onset of GAD for a number of reasons and patterns of comorbidity between GAD and PD have already been distinguished [26]. Underlying biological mechanisms such as inflammatory over-reactivity can be shared by medical diseases and anxiety disorders as in the case of cardiovascular diseases [8]. In this regard, some authors demonstrated that both GAD and PD patients presented higher interleukin (IL)-6 than healthy subjects and that PD patients had higher IL-6 plasma levels than GAD ones [28]. Moreover, while pre-treatment IL-6 levels negatively correlated with treatment response in PD in one study [29], other authors demonstrated that higher IL-6 levels were related to a better response to escitalopram treatment in PD [28]. In addition, worries linked to the prognosis of medical conditions (e.g., cancer) can favor the onset of GAD in biologically predisposed subjects [30]. Finally, sub-threshold somatic anxiety concerning a specific organ/system may trigger medical conditions as happens for example for atopic dermatitis [31] or irritable bowel syndrome [32]. Similarly, chronic dysfunction in different biological systems coupled with an unhealthy lifestyle (e.g., unbalanced diet or absence of physical activity) may increase the risk of different medical conditions in GAD patients [33,34]. On the other hand, the resistance of patients with GAD to seeking medical help, and the consequent progressive social deterioration and biological alterations shared by affective disorders predispose to the development of comorbid psychiatric disorders, particularly unipolar depression.

Third, regarding psychopharmacological treatment, we found that PD patients were largely treated with SSRIs, while GAD patients with SGAs (in particular quetiapine) and GABAergic drugs. These results are in line with available literature [35,36,37] and reflect the fact that only 50% of patients affected by GAD respond to a first-line treatment with SSRIs or selective serotonin-noradrenalin reuptake inhibitors (SNRIs) [22]. This aspect implies that most subjects affected by GAD are treated with second-line options or combined treatments [38]. Furthermore, we found that GAD patients discontinued treatment more frequently than those affected by PD as a result of poor compliance and re-exacerbations in line with literature about the path of GAD [17,39,40,41,42]. Patients affected by anxiety disorders and particularly GAD frequently report a fear of pharmacotherapy side effects; so, the choice of a specific compound should take into account its tolerability profile and speed of action. These are essential elements to obtain a good adherence to medical prescriptions [43,44].

Finally, patients affected by GAD were more frequently treated with psychodynamic psychotherapy than subjects suffering from PD. Psychodynamic psychotherapy is an insight-oriented form of therapy that aims to resolve unconscious conflicts linked to early-life relationships by using techniques such as clarifications, interpretations and confrontations [45]. This result is an original finding since most of the available literature highlights the efficacy of CBT for anxiety disorders [35,45,46,47], although evidence of the effectiveness of intensive short-term dynamic psychotherapy is emerging [48,49]. There are only a few studies comparing CBT and other techniques in the literature, suggesting more effectiveness of the first approach in GAD patients [50,51,52,53]. In addition, some reports showed that combining medication with psychotherapy may be more effective for patients with moderate to severe anxiety disorders with respect to pharmacotherapy alone [35,54]. In our sample, the patients with GAD might have been more preferentially referred to psychotherapy, as a result of less response to pharmacotherapy, in comparison with patients affected by PD. Alternatively, patients suffering from GAD may have simply preferred the psychodynamic psychotherapy approach of all the available therapeutic options. A review underlined how people with mental disorders who received their preferred treatment may have lower dropout rates and a higher therapeutic alliance [55], in line with the recent notion of a multi-disciplinary approach for the management of mental disorders, including a combination of pharmacologic and behavioral strategies [56]. Another plausible hypothesis is that there is a need to face and potentially solve problems related to an insecure attachment. Childhood trauma and neglect are early life stressors potentially underlying the increased risk of developing GAD or PD [57,58,59]. Moreover, insecure attachment, as a consequence of adverse childhood experiences, is also associated with anxiety disorders, especially GAD and emotional dysregulation [60]. While angry-dismissive style was associated only with GAD onset during adulthood and attachment style was unrelated to the development of PD [61], more recently it has been demonstrated that higher levels of subscales reflecting maternal insecure avoidant attachment (e.g., no memory of early childhood experiences and balancing/forgiving current state of mind) were more predictive of GAD relative to PD during adulthood [18].

The study presents several limitations. One main aspect is missing the psychosocial factors that may play a role in the enteropathogenesis of both anxiety disorders. Other limits related to the retrospective naturalistic design of the study are that (1) the sample size was relatively small; (2) the data were collected in two hospitals from the same geographic area so that the findings of the present article cannot be totally representative of other contexts; (3) diagnoses of anxiety disorders were made by a team of various psychiatrists; (4) no specific rating scales were used to assess symptom severity and side effects of pharmacological treatments. Moreover, no subjective questionnaires regarding satisfaction or quality of life were provided to patients, as reported by other studies [62]. Finally, a large part of the data are based on self-report, increasing the risk of recall bias and skewed self-presentation.

## 5. Conclusions

This study investigated the clinical differences between GAD and PD to optimize the management of patients with anxiety disorders in a framework of precision medicine. Our results seem to suggest a worse prognosis in GAD versus PD patients. More frequent multiple co-morbidities and relapses as well as poor compliance to treatment are aspects that seem to characterize patients affected by GAD, with respect to those suffering from PD. Further studies on larger samples are necessary to confirm our findings and shed light on the clinical differences between GAD and PD to ameliorate clinical practice and patients’ prognosis.

## Figures and Tables

**Table 1 jpm-13-00491-t001:** Descriptive and univariate analyses: comparison between GAD and PD in terms of sociodemographic and clinical variables.

Variables		GAD N = 131	PD N = 159	Total Sample N = 290	F or χ^2^	*p* Value (OR)
Age (years)		48.2 (±15.6)	42.4 (±14.6)	45.1 (±15.4)	10.37	**<0.01**
Age at onset (years)		40.0 (±16.0)	35.6 (±14.4)	37.6 (±15.3)	5.92	**0.02**
Duration of illness (months)		106.4 (±116.5)	85.5 (±88.8)	95.8 (±103.1)	2.96	0.09
DUI (months)		39.7 (±77.0)	27.3 (±50.6)	32.7 (±63.8)	2.66	0.10
Treatment duration (months)		42.3 (±60.4)	37.8 (±61.7)	40.6 (±62.8)	0.38	0.54
Gender	Male	40 (30.5%)	65 (40.9%)	105 (36.2%)	3.33	0.08 (0.64)
	Female	91 (69.5%)	94 (59.1%)	185 (63.8%)		
Family history of psychiatric disorders	None	72 (55.0%)	81 (50.9%)	153 (52.8%)	12.83	0.15
	SKZ	9 (6.9%)	5 (3.1%)	14 (4.8%)		
	BD	8 (6.1%)	6 (3.8%)	14 (4.8%)		
	Unipolar depression	25 (19.1%)	30 (18.9%)	55 (19.0%)		
	GAD	12 (9.2%)	24 (15.1%)	36 (12.4%)		
	PD	1 (0.7%)	5 (3.1%)	6 (2.1%)		
	Social phobia	0 (0.0%)	3 (1.9%)	3 (1.0%)		
	OCD	0 (0.0%)	2 (1.3%)	2 (0.7%)		
	Eating disorder	1 (0.7%)	2 (1.3%)	3 (1.0%)		
	Substance abuse	3 (2.3%)	1 (0.6%)	4 (1.4%)		
Multiple family history of psychiatric disorders	No	117 (89.3%)	135 (84.9%)	252 (86.9%)	1.22	0.30 (1.49)
	Yes	14 (10.7%)	24 (15.1%)	38 (13.1%)		
Work status	Employed	110 (84.0%)	123 (77.4%)	233 (80.3%)	1.99	0.18 (0.65)
	Unemployed	21 (16.0%)	36 (22.6%)	57 (19.7%)		
Marital Status	Single	40 (30.5%)	65 (40.9%)	105 (36.2%)	3.34	0.20
	Married	77 (58.8%)	80 (50.3%)	157 (54.1%)		
	Divorced	14 (10.7%)	14 (8.8%)	28 (9.7%)		
Pre-onset psychiatric comorbidity	None	92 (70.2%)	122 (76.7%)	214 (73.8%)	13.85	0.07
	Unipolar depression	18 (13.7%)	14 (8.8%)	32 (11.0%)		
	GAD	2 (1.5%) ^●^	9 (5.7%)	11 (3.9%)		
	PD	8 (6.1%)	2 (1.3%) ^●^	10 (3.4%)		
	Social phobia	1 (0.8%)	3 (1.9%)	4 (1.4%)		
	OCD	2 (1.5%)	1 (0.6%)	3 (1.0%)		
	PTSD	1 (0.8%)	2 (1.2%)	3 (1.0%)		
	Eating disorders	6 (4.6%)	3 (1.9%)	9 (3.1%)		
	Others (ADHD/BD/Hypochondria/SLD)	1 (0.8%)	3 (1.9%)	4 (1.4%)		
Pre-onset psychiatric multiple co-morbidities	No	121 (92.4%)	151 (95.0%)	272 (93.8%)	0.83	0.46 (0.64)
	Yes	10 (7.6%)	8 (5.0%)	18 (6.2%)		
Post-onset psychiatric comorbidity	None	83 (63.4%)	122 (76.7%)	205 (70.7%)	10.57	0.22
	Unipolar depression	28 (21.4%)	20 (12.7%)	48 (16.6%)		
	GAD	1 (0.8%) ^●^	3 (1.9%)	4 (1.4%)		
	PD	5 (3.8%)	6 (3.8%) ^●^	11 (3.8%)		
	Social phobia	2 (1.5%)	1 (0.6%)	3 (1.0%)		
	OCD	2 (1.5%)	1 (0.6%)	3 (1.0%)		
	PTSD	1 (0.8%)	1 (0.6%)	2 (0.7%)		
	Eating disorders	5 (3.8%)	1 (0.6%)	6 (2.1%)		
	Others (ADHD/BD/Hypochondria/SLD)	4 (3.0%)	4 (2.5%)	8 (2.7%)		
Post-onset psychiatric multiple co-morbidities	No	113 (86.3%)	150 (94.3%)	263 (90.7%)	5.55	**0.02** **(0.38)**
	Yes	18 (13.7%)	9 (5.7%)	27 (9.3%)		
Personality disorder	None	115 (87.7%)	149 (93.7%)	264 (91.0%)	8.20	0.74
	Schizoid	1 (0.8%)	1 (0.6%)	2 (0.7%)		
	Schizotypal	1 (0.8%)	0 (0.0%)	1 (0.3%)		
	Paranoid	1 (0.8%)	0 (0.0%)	1 (0.3%)		
	Histrionic	2 (1.5%)	0 (0.0%)	2 (0.7%)		
	Narcissistic	1 (0.8%)	0 (0.0%)	1 (0.3%)		
	Borderline	4 (3.0%)	5 (3.2%)	9 (3.2%)		
	Avoidant	1 (0.8%)	1 (0.6%)	2 (0.7%)		
	Obsessive-compulsive	2 (1.5%)	1 (0.6%)	3 (1.1%)		
	Dependent	1 (0.8%)	0 (0.0%)	1 (0.3%)		
	NOS	2 (1.5%)	2 (1.3%)	4 (1.4%)		
Pre-onset substance misuse	None	119 (90.7%)	141 (88.7%)	260 (89.7%)	4.54	0.66
	Alcohol	7 (5.3%)	6 (3.7%)	13 (4.3%)		
	Cannabis	1 (0.8%)	4 (2.5%)	5 (1.7%)		
	Cocaine	1 (0.8%)	4 (2.5%)	5 (1.7%)		
	Heroin	1 (0.8%)	2 (1.3%)	3 (1.0%)		
	Benzodiazepine	1 (0.8%)	2 (1.3%)	3 (1.0%)		
	Other	1 (0.8%)	0 (0.0%)	1 (0.3%)		
Pre-onset poly-substance misuse	No	126 (96.2%)	153 (96.2%)	279 (96.2%)	0.00	1.00 (0.99)
	Yes	5 (3.8%)	6 (3.8%)	11 (3.8%)		
Pre-onset substance misuse	None	121 (92.4%)	146 (91.8%)	267 (92.2%)	4.82	0.62
	Alcohol	6 (4.5%)	6 (3.9%)	12 (4.1%)		
	Cannabis	1 (0.8%)	4 (2.5%)	5 (1.8%)		
	Cocaine	0 (0.0%)	1 (0.6%)	1 (0.3%)		
	Heroin	0 (0.0%)	1 (0.6%)	1 (0.3%)		
	Benzodiazepine	2 (1.5%)	1 (0.6%)	3 (1.0%)		
	Other	1 (0.8%)	0 (0.0%)	1 (0.3%)		
Post-onset substance misuse	No	130 (99.2%)	155 (97.5%)	285 (98.3%)	1.30	0.38 (3.35)
	Yes	1 (0.8%)	4 (2.5%)	5 (1.7%)		
Pre-onset medical comorbidity	None	79 (60.3%)	108 (67.9%)	187 (64.5%)	9.64	0.68
	Obesity	4 (3.1%)	3 (1.9%)	7 (2.4%)		
	Hypercholesterolemia	4 (3.1%)	2 (1.3%)	6 (2.1%)		
	Hypertension	14 (10.7%)	13 (8.2%)	27 (9.3%)		
	Diabetes	3 (2.3%)	2 (1.3%)	5 (1.7%)		
	Hyperthyroidism	2 (1.5%)	2 (1.3%)	4 (1.4%)		
	Hypothyroidism	4 (3.1%)	4 (2.5%)	8 (2.8%)		
	Stroke	2 (1.5%)	0 (0.0%)	2 (0.7%)		
	Epilepsy	2 (1.5%)	1 (0.6%)	3 (1.0%)		
	Migraine	5 (3.8%)	4 (2.5%)	9 (3.1%)		
	Gastrointestinal diseases	8 (6.1%)	8 (5.0%)	16 (5.5%)		
	Cancer	2 (1.5%)	8 (5.0%)	10 (3.4%)		
	Other (AMI or Asthma)	2 (1.5%)	4 (2.5%)	6 (2.1%)		
Pre-onset medical multiple co-morbidities	No	99 (75.6%)	138 (86.8%)	237 (81.7%)	6.05	**0.01** **(0.47)**
	Yes	32 (24.4%)	21 (13.2%)	53 (18.3%)		
Post-onset medical comorbidity	None	70 (53.4%)	109 (68.7%)	179 (61.8%)	11.90	0.46
	Obesity	5 (3.8%)	3 (1.9%)	8 (2.8%)		
	Hypercholesterolemia	5 (3.8%)	3 (1.9%)	8 (2.8%)		
	Hypertension	19 (14.5%)	11 (6.9%)	30 (10.3%)		
	Diabetes	4 (3.0%)	3 (1.9%)	7 (2.4%)		
	Hyperthyroidism	1 (0.8%)	0 (0.0%)	1 (0.3%)		
	Hypothyroidism	6 (4.6%)	5 (3.1%)	11 (3.8%)		
	Stroke	0 (0.0%)	1 (0.6%)	1 (0.3%)		
	Epilepsy	1 (0.8%)	1 (0.6%)	2 (0.7%)		
	Migraine	6 (4.6%)	5 (3.1%)	11 (3.8%)		
	Gastrointestinal diseases	8 (6.2%)	10 (6.3%)	18 (6.2%)		
	Cancer	4 (3.0%)	5 (3.1%)	9 (3.1%)		
	Other (AMI or Asthma)	2 (1.5%)	3 (1.9%)	5 (1.7%)		
Post-onset medical multiple co-morbidities	No	92 (70.2%)	138 (86.8%)	230 (79.3%)	12.01	**<0.01** **(0.36)**
	Yes	39 (29.8%)	21 (13.2%)	60 (20.7%)		
Suicide attempts	No	127 (96.9%)	158 (99.4%)	285 (98.3%)	2.49	0.18 (0.20)
	Yes	4 (3.1%)	1 (0.6%)	5 (1.7%)		
Hospitalizations	No	115 (87.8%)	148 (93.1%)	263 (90.7%)	2.38	0.15 (0.53)
	Yes	16 (12.2%)	11 (6.9%)	27 (9.3%)		
Obstetric complications	No	129 (98.5%)	158 (99.4%)	287 (99.0%)	0.56	0.59 (0.41)
	Yes	2 (1.5%)	1 (0.6%)	3 (1.0%)		
Type of obstetric complications	None	129 (98.6%)	158 (99.4%)	287 (99.1%)	2.43	0.21
	Low birth weight (<2.5 kg)	1 (0.7%)	0 (0.0%)	1 (0.3%)		
	Preterm (<32 weeks)	1 (0.7%)	0 (0.0%)	1 (0.3%)		
	Unknown	0 (0.0%)	1 (0.6%)	1 (0.3%)		
Treatment	None	11 (8.4%)	10 (6.2%)	21 (7.2%)	19.09	**0.01**
	Multimodal antidepressant *	4 (3.0%)	4 (2.5%)	8 2.8%)		
	SNRI	9 (6.9%)	9 (5.7%)	18 (6.2%)		
	SSRI	77 (58.8%)	118 (74.2%)	195 (67.2%)		
	TCA	3 (2.3%)	3 (1.9%)	6 (2.1%)		
	Other ADs ⁰	2 (1.5%)	4 (2.5%)	6 (2.1%)		
	GABAergic drugs	9 (6.9%)	1 (0.6%)	10 (3.4%)		
	SGAs	8 (6.1%)	2 (1.3%)	10 (3.4%)		
	Mood stabilizer ^■^	0 (0.0%)	2 (1.3%)	2 (0.7%)		
	Benzodiazepines	8 (6.1%)	6 (3.8%)	14 (4.9%)		
Poly-therapy	No	52 (39.7%)	71 (44.7%)	123 (42.4%)	0.72	0.41 (0.82)
	Yes	79 (60.3%)	88 (55.3%)	167 (57.6%)		
Side effects	None	117 (89.3%)	137 (86.1%)	254 (87.6%)	5.15	0.70
	Gain weight	4 (3.0%)	6 (3.8%)	10 (3.4%)		
	Somnolence	5 (3.8%)	3 (1.9%)	8 (2.8%)		
	Sexual dysfunction	2 (1.5%)	6 (3.8%)	8 (2.8%)		
	Motor disturbances	0 (0.0%)	3 (1.9%)	3 (1.0%)		
	Nausea	1 (0.8%)	1 (0.6%)	2 (0.7%)		
	Tension	1 (0.8%)	1 (0.6%)	2 (0.7%)		
	Others **	1 (0.8%)	2 (1.3%)	3 (1.0%)		
Multiple side effects	No	128 (97.7%)	154 (96.9%)	282 (97.2%)	0.20	0.73 (1.38)
	Yes	3 (2.3%)	5 (3.1%)	8 (2.8%)		
Reason for discontinuation	No discontinuation ^⌂^	113 (86.3%)	148 (93.1%)	261 (90.1%)	10.10	**0.01**
	Side effects	2 (1.5%)	5 (3.1%)	7 (2.4%)		
	Relapse/Hospitalizations	9 (6.9%)	1 (0.7%)	10 (2.4%)		
	No compliance	7 (5.3%)	5 (3.1%)	12 (4.1%)		
Lifetime psychotherapy	No	81 (61.8%)	108 (67.9%)	189 (65.2%)	1.17	0.32 (0.76)
	Yes	50 (38.2%)	51 (32.1%)	101 (34.8%)		
Type of psychotherapy	None	81 (61.8%)	107 (67.3%)	188 (64.8%)	8.13	**0.04**
	Psychoeducational/ Supportive	8 (6.1%)	7 (4.4%)	15 (5.2%)		
	CBT	24 (18.3%)	24 (15.1%)	48 (16.6%)		
	Psychodynamic	12 (9.2%)	3 (1.9%)	15 (5.2%)		
	Unknown	6 (4.6%)	18 (11.3%)	24 (8.2%)		

Legend: * vortioxetine; ⁰ trazodone, agomelatine, mirtazapine; ^■^ valproic acid or lithium; ** tachycardia, dizziness, headache, constipation, diarrhea; ^⌂^ patients who discontinued for long-term treatment effectiveness are included in this group; ^●^ patients who had previously another main diagnosis; AD, antidepressant; ADHD, attention deficit hyperactivity disorder; AMI, acute myocardial infarction; BD, bipolar disorder; CBT, cognitive behavioral therapy; DUI, duration of untreated illness; GAD, generalized anxiety disorder; MDMA, methylenedioxymethamphetamine; NOS, not otherwise specified; OCD, obsessive compulsive disorder; OR, odds ratio (GAD/PD); PD, panic disorder; PTSD, post-traumatic stress disorder; SGA, second generation antipsychotic; SKZ, schizophrenia; SLD, specific learning disorder; SNRI, serotonin and noradrenalin reuptake inhibitor; SSRI, selective serotonin reuptake inhibitor; TCA, tricyclic antidepressant. Mean (±standard deviation) for quantitative variables; frequencies (percentage) for qualitative variables. In **bold** statistically significant differences (*p* < 0.05).

**Table 2 jpm-13-00491-t002:** Association between significant variables and GAD/PD diagnosis: logistic regression model analysis.

Variables	*p* Value	OR	95% C.I.	
			Inferior	Superior
Age	0.20	0.98	0.94	1.01
Age at onset	0.94	1.00	0.96	1.03
Post-onset psychiatric multiple co-morbidities	**0.05**	**0.38**	0.16	0.87
Pre-onset medical multiple co-morbidities	0.13	0.47	0.26	0.86
Post-onset medical multiple co-morbidities	**0.02**	**0.36**	0.20	0.65
Main treatment	**0.03**	NA	NA	NA
Type of psychotherapy	**<0.01**	NA	NA	NA
Reason for discontinuation	**0.02**	NA	NA	NA

Legend: B, regression coefficient; C.I., confidence interval; df, degrees of freedom; NA, not applicable; OR, odds ratio; S.E., standard error. In **bold** statistically significant differences (*p* < 0.05).

## Data Availability

Data sharing is not applicable to this article.

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
