# Peer review of "Clinical Markers of Panic and Generalized Anxiety Disorder: Overlapping Symptoms, Different Course and Outcome"

_jpm, 2023, doi:10.3390/jpm13030491_

Round 1
Reviewer 1 Report
The work is not of any great scientific significance for the rank M22, because there are no new conclusions.
The statistics are done correctly.
Иn table number 2 it is not necessary to write b, Wald, df and S.E.
You have OR, 95% C .I. and p value.
Author Response
First of all, we would like to thank the First Reviewer for the interest in our manuscript and the useful comments to improve the paper.
As suggested, we removed b, Wald, df and S.E. from Table 2.
Reviewer 2 Report
The theme of clinical markers of Panic and Generalized Anxiety Disorder and the evaluation of the associations between clinical/socio-demographic markers and GD/PT diagnosis has a practical relevance and importance.
Even if the theme is interesting, I found this paper at too „one-dimensional“ and biologically centred.
I have some remarks and suggestions how to improve the paper. Although the factors I mention below were not examined in the study, authors could mention them in the discussion and in the limitations.
Authors did not mention important clinical features as inflammatory factors - IL-6 or IFN-γ, with differences in GAD and PD.
Zou, Z., Zhou, B., Huang, Y., Wang, J., Min, W., & Li, T. (2020). Differences in cytokines between patients with generalised anxiety disorder and panic disorder. Journal of Psychosomatic Research, 133, 109975
Already the older paper of Russel from 1992 differentiates PD subjects (with autonomic hyperactivity) from GAD subjects (with central nervous system hyperarousal).
RUSSELL NOYES, J., Woodman, C., Garvey, M. J., Cook, B. L., Suelzer, M., Clancy, J., & Anderson, D. J. (1992). Generalized anxiety disorder vs. panic disorder: Distinguishing characteristics and patterns of comorbidity. The Journal of nervous and mental disease, 180(6), 369-379.
The paper omits psychosocial factors and underlying mechanisms of anxiety symptoms (such as neuroendocrine and immunological changes associated with early stress – e.g childhood abuse and neglect).
Heim, C., & Nemeroff, C. B. (2001). The role of childhood trauma in the neurobiology of mood and anxiety disorders: preclinical and clinical studies. Biological psychiatry, 49(12), 1023-1039.
Kuzminskaite, E., Penninx, B. W., van Harmelen, A. L., Elzinga, B. M., Hovens, J. G., & Vinkers, C. H. (2021). Childhood trauma in adult depressive and anxiety disorders: an integrated review on psychological and biological mechanisms in the NESDA cohort. Journal of affective disorders, 283, 179-191.
As I mentioned before, the paper omits psychosocial factors (abuse and neglect in the childhood and insecure attachment) that might play a role in the etiopathogenesis of both anxiety disorders.
Safren, S. A., Gershuny, B. S., Marzol, P., Otto, M. W., & Pollack, M. H. (2002). History of childhood abuse in panic disorder, social phobia, and generalized anxiety disorder. The Journal of nervous and mental disease, 190(7), 453-456.
The insecure attachment (related to adverse childhood experiences) is also associated with anxiety disorders, especially with GAD and with emotional dysregulation.
Marganska, A., Gallagher, M., & Miranda, R. (2013). Adult attachment, emotion dysregulation, and symptoms of depression and generalized anxiety disorder. American journal of orthopsychiatry, 83(1), 131
Higher levels of subscales reflecting maternal insecure avoidant attachment (e.g., no memory of early childhood experiences and balancing/forgiving current state of mind) emerged as more predictive of GAD relative to PD.
Newman, M. G., Shin, K. E., & Zuellig, A. R. (2016). Developmental risk factors in generalized anxiety disorder and panic disorder. Journal of affective disorders, 206, 94-102.
Another study found that angry-dismissive style was associated only with GAD and attachment style was unrelated to panic disorder.
Bifulco, A., Kwon, J., Jacobs, C., Moran, P. M., Bunn, A., & Beer, N. (2006). Adult attachment style as mediator between childhood neglect/abuse and adult depression and anxiety. Social psychiatry and psychiatric epidemiology, 41, 796-805.
Interestingly, the association between GAD and psychodynamic psychotherapy as an original finding of your study can clearly point to the fact that there is a need (probably seen by clinicians as well as by patients) to solve the problems associated with insecure attachment.
Author Response
First of all, we would like to thank the Second Reviewer for the interest in our manuscript and the useful comments to improve the paper.
As suggested, we added the requested information and references in the discussion and limits.
Round 2
Reviewer 2 Report
Dear Authors,
Thank you for providing corrections and improvements of the paper. It is now little more "balanced" even if it is too biologically centred.
Your reviewer
Author Response
Dear reviewer,
thank you very much for your useful comments. In a future work we are planning to enlarge the sample and to analyse also psychological factors for GAD and PD as you suggested.
Thank you
Best regards